# Discovery of Novel Cyclic Ethers with Synergistic Antiplasmodial Activity in Combination with Valinomycin

**DOI:** 10.3390/molecules26247494

**Published:** 2021-12-10

**Authors:** Daniel J. Watson, Paul R. Meyers, Kojo Sekyi Acquah, Godwin A. Dziwornu, Christopher Bevan Barnett, Lubbe Wiesner

**Affiliations:** 1Division of Clinical Pharmacology, Department of Medicine, University of Cape Town, Cape Town 7700, South Africa; lubbe.wiesner@uct.ac.za; 2Department of Molecular and Cell Biology, University of Cape Town, Cape Town 7700, South Africa; paul.meyers@uct.ac.za (P.R.M.); ACQKOJ001@myuct.ac.za (K.S.A.); 3Department of Chemistry, University of Cape Town, Cape Town 7700, South Africa; godwin.dziwornu@uct.ac.za (G.A.D.); chris.barnett@uct.ac.za (C.B.B.)

**Keywords:** malaria, ionophores, cyclic ethers, polypropylene glycol, valinomycin, synergy

## Abstract

With drug resistance threatening our first line antimalarial treatments, novel chemotherapeutics need to be developed. Ionophores have garnered interest as novel antimalarials due to their theorized ability to target unique systems found in the *Plasmodium*-infected erythrocyte. In this study, during the bioassay-guided fractionation of the crude extract of *Streptomyces* strain PR3, a group of cyclodepsipeptides, including valinomycin, and a novel class of cyclic ethers were identified and elucidated. Further study revealed that the ethers were cyclic polypropylene glycol (cPPG) oligomers that had leached into the bacterial culture from an extraction resin. Molecular dynamics analysis suggests that these ethers are able to bind cations such as K^+^, NH_4_^+^ and Na^+^. Combination studies using the fixed ratio isobologram method revealed that the cPPGs synergistically improved the antiplasmodial activity of valinomycin and reduced its cytotoxicity *in vitro*. The IC_50_ of valinomycin against *P. falciparum* NF54 improved by 4–5-fold when valinomycin was combined with the cPPGs. Precisely, it was improved from 3.75 ± 0.77 ng/mL to 0.90 ± 0.2 ng/mL and 0.75 ± 0.08 ng/mL when dosed in the fixed ratios of 3:2 and 2:3 of valinomycin to cPPGs, respectively. Each fixed ratio combination displayed cytotoxicity (IC_50_) against the Chinese Hamster Ovary cell line of 57–65 µg/mL, which was lower than that of valinomycin (12.4 µg/mL). These results indicate that combinations with these novel ethers may be useful in repurposing valinomycin into a suitable and effective antimalarial.

## 1. Introduction

Ionophores are ion carriers that can selectively bind cations and transport them down electrochemical gradients [1]. Ionophores typically have a hydrophilic interior in which they coordinate cations, and a hydrophobic exterior which allows them to transport cations through lipid membranes. This ability allows ionophores to disrupt electrochemical gradients in cells, resulting in cellular disruption and death [2]. Due to this, ionophores exhibit a range of antibiotic activities, including antibacterial [1], antifungal [3], anticancer [4] and antiviral [5]. However, most ionophores are barred from being used as chemotherapeutic agents in humans due to their lack of host selectivity [2]. Despite this, ionophores are gathering interest as novel antimalarial drugs, with a number of studies showing that ionophores display potent antimalarial activity *in vitro* and *in vivo* [2,6,7,8,9,10].

Malaria is a severe tropical disease caused by five parasites of the *Plasmodium* genus, with *Plasmodium falciparum* reported to be responsible for most infections and mortalities. In 2019, the World Health Organization (WHO) estimated that malaria was responsible for 409,000 deaths and 229 million new infections [11]. The artemisinin-based combination therapies (ACT) are the WHO-recommended first line treatments for *Plasmodium* infection. However, the continued use of ACTs is currently threatened by the development of drug resistance and, to continue combating malaria, novel treatments with novel mechanisms of action need to be developed [12,13]. During infection of the erythrocyte, *P. falciparum* completely remodels the host’s membrane and transport systems [6,14]. Part of this remodeling involves a large-scale increase in ion transport capacity, such as increases in K^+^, Na^+^, and Cl^−^ transport systems and the inhibition of erythrocyte ATPase membrane pumps [15]. This creates a unique environment in *P. falciparum*-infected erythrocytes, the formation of which could be selectively inhibited by ionophores [2,15]. Therefore, more study is warranted on ionophores that could selectively target the *P. falciparum* pathogen.

Members of the bacterial genus *Streptomyces* are known to be prolific producers of antibiotic compounds, including ionophores. Examples include salinomycin, nigericin, narasin and lasalocid, each of which have displayed antiplasmodial activity *in vitro* [2,4,6]. In this study, during the bioassay-guided fractionation of *Streptomyces* strain PR3 culture broths, two series of antiplasmodial compounds were isolated and studied, one produced by *Streptomyces* strain PR3 and one isolated from the XAD-16N resin used during extraction of the culture cell mass.

## 2. Results and Discussion

### 2.1. Identification of Cyclodepsipeptides

*Streptomyces* strain PR3 was identified as a producer of potent antiplasmodial compounds during a series of experiments to screen filamentous actinobacteria [16]. In this study, to isolate and identify the active compounds being produced, a typical bioassay-guided fractionation study was undertaken. Ten liters of strain PR3 were cultured and extracted with Amberlite XAD-16N resin. The ethyl acetate (EtAc) extract of the cell mass and Amberlite XAD-16N resin were collected, dried, and fractionated using two rounds of solid phase extraction (SPE). Each fraction was tested for antiplasmodial activity against the drug-sensitive strain *P. falciparum*, NF54. The methanol (MeOH) wash step (fraction #3, Table 1, Appendix A) maintained antiplasmodial activity and was subsequently fractionated by a normal phase SPE system to produce 5 active fractions (fractions #5, #6, #7, #8 and #13, Table 1, Appendix A).

Active fractions were analyzed by high performance liquid chromatography–high resolution electrospray ionization mass spectrometry (HPLC–HRESIMS) and processed using Global Natural Products Social (GNPS) molecular networking (http://gnps.ucsd.edu, 15 July 2021) [17]. A series of known and novel cyclodepsipeptides were identified in fractions #5–#8 (Figure 1, Appendix A). These results were supported by two dereplication software programs, Dereplicator and Varquest, and by comparison to the literature [18,19,20,21,22,23]. The GNPS molecular networking, Dereplicator and Varquest jobs are provided in the Appendix A. Valinomycin was detected at *m*/*z* 1111.6409 [M + H]^+^ and 1128.6616 [M + NH_4_]^+^. Valinomycin is a neutral ionophore and coordinates monovalent cations by ion-dipole interactions between its carboxylic oxygen atoms in its inner ring [1]. Valinomycin displays a wide range of antibiotic properties, including antiplasmodial activity [7]. Other ionophores were detected in fractions #5–#8, including montanastatin at *m*/*z* 741.4288 [M + H]^+^ and *m*/*z* 758.4552 [M+NH_4_]^+^, four recently elucidated cyclodepsipeptides, Streptodepsipeptide P11B at *m*/*z* 1100.6308 [M + NH_4_]^+^, Streptodepsipeptide P11A at *m*/*z* 1114.6461 [M + NH_4_]^+^, Streptodepsipeptide SV21 at *m*/*z* 1142.6778 [M + NH_4_]^+^ and an unnamed cyclodepsipeptide at *m*/*z* 1156.6997 [M + NH_4_]^+^ [18,23]. Two nodes with *m*/*z* 1072.5997 and 1086.6151 were detected with the other cyclodepsipeptides, which are yet to be identified. As they are clustered together and differ by small masses from the other molecules, they too are likely analogues of valinomycin. The total ion chromatograms of fractions #5–#8 indicate that valinomycin is present in the highest intensity, followed by the novel cyclodepsipeptides (Appendix A). Taking this and its known antiplasmodial activity into account, it is likely that valinomycin is responsible for the activity displayed by these fractions.

### 2.2. Elucidation of the Cyclic Polypropylene Glycol

The molecular network of fraction #13 revealed no known compounds. The dominant signal was an ammonium adduct with an *m*/*z* of 888.6597 [M+NH_4_]^+^; a protonated form with *m*/*z* 871.6373 [M+H]^+^ was also detected (Figure 2). Henceforth, this molecule will be referred to as compound 870. The isotope pattern of compound 870 suggested it has a carbon backbone (Appendix A). Comparison of the precursor masses to the SciFinder, KEGG and MassBank libraries revealed no matches. Closer analysis of compound 870’s MS/MS spectrum revealed a consistent mass shift of 58.042 Da (Figure 2). From analyses of the proton (^1^H), carbon (^13^C) and heteronuclear single quantum coherence (HSQC) NMR spectra, signals corresponding to a methyl (δ_H_ 1.30, δ_C_ 16.2), methylene (δ_H_ 3.48 and 3.55, δ_C_ 72.9), and methine group (δ_H_ 3.63, δ_C_ 75.3) were detected (Table 2, Figure 3 and Appendix A). Comparing the mass of this compound and the very few corresponding ^1^H and ^13^C NMR signals, the compound was found to either have a repeating motif, which is typical of polymers, or some form of symmetry. The correlation spectroscopy (^1^H-^1^H COSY) spectrum revealed only a single spin system between the methine, diastereotopic methylene and methyl protons (Appendix A), which was further confirmed by analysis of the heteronuclear multiple-bond coherence (HMBC) NMR spectrum (Appendix A). The deshielded ^1^H and ^13^C NMR chemical shifts of the methylene and methine groups indicated they were potentially attached to the electronegative oxygen atom, as corroborated by the HRESIMS and consistent with a propylene glycol monomer (C_3_H_6_O, 58.041 Da). As 888.6597 is an ammonium adduct and 871.6373 is protonated, their experimental mass is 870.6271 Da. Further, 870.6271 is wholly divisible by 58.042 to produce 15 units, suggesting that the polymer is cyclic and comprised of 15 propylene glycol monomers, which form a cyclic oligoether (Figure 4 and Figure 5).

Analysis of fraction #13 by GNPS molecular networking revealed the presence of a series of cyclic polypropylene glycols (cPPGs) (Figure 5). Each signal is an ammonium adduct and differs by 58.042 Da, which represents a propylene glycol monomer. These molecules have an empirical formula of C_3n_H_6n_O_n_, where n is the number of oxygens in the backbone (Appendix A).

Originally, it was believed that *Streptomyces* strain PR3 was producing the cPPGs. However, the MS and MS/MS spectra of the cPPGs have been reported in previous literature. An investigation on the leaching of plastic oligomers in medical implants reported a series of PPG oligomers in isopropanol alcohol extracts [24]. The mass spectra of these reported PPG oligomers matched the mass spectra of the cPPGs. However, the structures of the PPG oligomers were not elucidated. This strongly suggested that the cPPGs are in fact xenobiotics that have leached into the bacterial cultures from an external source.

In this study, to determine the source of the cPPGs, each component of the crude extraction was washed with EtAc and water, the two solvents used in the extraction process. This included the glassware used, the culture media ingredients, the XAD-16N extraction resin and the plastic benchtop centrifuge tubes. The solvents were dried down and analyzed by HPLC–HRESIMS and the GNPS molecular networking platform. Only the XAD-16N sample contained cPPGs (Appendix A).

Interestingly, cPPGs were only detected when the XAD16-N extraction resin was washed with water and EtAc, and could not be observed with organic solvent alone. This suggests that water plays a key part in the cyclization of the PPG oligomers. The cyclization of plastic oligomers into macrocycles has been reported before. Polylactic acid has been shown to form cyclic oligomers between 2–14 monomers in length [25]. This evidence suggests that when the XAD-16N resin was added to the culture prior to extraction, PPG oligomers present in the resin cyclized into the cPPGs, then bound to the resin and were extracted during the EtAc wash step of the crude extraction.

The cPPGs displayed moderate antiplasmodial activity, with a mean IC_50_ of 2104 ng/mL recorded against *P. falciparum* (Table 1). Polyethers, such as lonomycin A, nigericin and monensin, are known to possess antiplasmodial activity [6,7]. However, not much is understood about the antiplasmodial activity of macrocyclic polyethers such as cPPGs.

cPPGs resemble crown ethers in structure, which are known to possess antibacterial and anticancer properties [26,27]. However, to be classed as crown ethers, cPPGs must exhibit ionophoric behaviour [1].

## 3. Computational Studies

MD calculations were conducted in order to investigate the conformational space of a selected cPPG (compound 870) and to determine if it can arrange to bind cations [28]. Radial distribution functions (RDF), a type of particle density histogram, showed that the cations (sodium, potassium, and ammonium ions) were within a close radius of the macrocycle (Figure 6). These unexpected short-range orderings were the first evidence that the ions were closely associating with the macrocycle. No long-range ordering was observed, as expected in a liquid. The first maximum in the RDF was at approximately 2.5 Å, indicating close association between each ion and the macrocycle (Figure 6). In potassium chloride solution, the density of K^+^ cations in this first ‘solvation’ shell of the macrocycle was greater than that found for the other cations.

The RDFs indicated that there were relevant close contacts to investigate but did not prescribe these as interesting bound states. MDAnalysis was used to select these structures, and these were visualized in VMD [29] (Figure 7). Visual analysis showed that the macrocycles can take on multiple conformations (circular, obloid, twisted, self-associated), including conformers where sections of the macrocycle arrange to form a crown ether motif that can bind cations.

The root mean square distance (RMSD) and the root mean square fluctuation (RMSF) calculations confirmed that the macrocycle is quite flexible and can take on a variety of conformations. In a sodium chloride waterbox, the RMSD of the macrocycle was an average of 4 Å and the radius of gyration an average of 6 Å; these values were similar for all systems. The RMSF was high, on average 25 Å and similar for all systems, indicating large fluctuations of heavy atoms positions and conformational flexibility. The RMSF calculation was calculated for the heavy atoms of the macrocycle, and a plot of the RMSF showed a different behavior for each system (Appendix A). This is likely related to the mode of binding for each ion.

Bidentate and crown ether type motifs with the ion inside the macrocycle were considered of interest, while nearby association was not considered sufficiently interesting. A few selected structures from the simulations are shown in Figure 7, with the bidentate and crown ether binding motifs noted. Further studies considering binding free energy may be helpful in predicting the binding behavior of each ion. However, MD simulations have shown that the cPPG macrocycle is able to access conformers that allow ion binding, which agrees with the experimental observations.

To further understand the range of conformers available to compound 870, a clustering analysis was conducted using TTClust [30]. The clusters were chosen automatically based on all atoms of the macrocycle and clustered using the Ward algorithm. This analysis yielded five clusters for each system (Appendix A) and confirmed that representative conformations from the simulations include but are not limited to circular, obloid, twisted, self-associated and crown-ether-like conformations.

These MD simulations illustrate that compound 870 can organize itself into conformations that are able to capture ions, which suggests that cPPGs may be a novel class of crown ethers. However, more extensive study is required to confirm this.

## 4. Combination Studies

During the bioassay-guided fractionation of cultures of strain PR3, active crude extracts were fractionated by SPE using C18 cartridges, then the active fraction (#3) was fractionated by normal phase SPE (Appendix A). Fraction #3 displayed comparable antiplasmodial activity to the purer normal phase fraction #6, with IC_50_s of 40.0 ± 1.5 ng/mL vs. 31.3 ± 4.2 recorded, respectively (Table 1). Typically in bioassay-guided fractionation, activity improves as the samples are purified [31]. However, synergy between components in mixtures is well-known and can lead to a stronger or equal response in much cruder fractions than pure samples [31,32]. Both identified antiplasmodial agents (cyclodepsipeptides and cPPGs) are ionophores, and antimalarial synergy has been documented for this drug class. For example, monensin and nigericin had a synergistic effect when tested in combination [33]. Therefore, it was hypothesized herein that the cPPGs and cyclodepsipeptides were interacting, and further investigation was undertaken. Fraction #6 (cyclodepsipeptides) was tested in combination with fraction #13 (cPPGs) to investigate this relationship. Fraction #6 was selected as it was the most active fraction of the fractions containing the cyclodepsipeptides.

Currently, there is no standardized methodology to determine the relationship between compounds [34]. The fixed-ratio isobologram method, based on the Loewe additivity model, is the most popular for studying combination effects in many fields, including antimalarial drug research, and was selected to determine the relationship between the cyclodepsipeptides and the cPPGs [34,35,36]. The fractional IC_50_ (FIC_50_) is calculated for each drug combined in a series of volumetric fixed ratios, and the nature of the interaction between the two compounds is determined by the sum of the fractional IC_50_s (ƩFIC) and the shape of the corresponding isobologram [37].

When plotted as an isobologram, the FIC_50_s of the cyclodepsipeptides and the cPPGs fell below the additivity line and had a general concave curve, suggesting synergy (Figure 8) [35]. For this investigation, the relationship between the compounds was classed as synergistic if the ƩFIC was ≤0.5 [32,37] (Appendix A). When the cyclodepsipeptides and the cPPGs were combined in the fixed ratios of 4:1, 3:2 and 1:4, the results suggest that there was an additive, or weak synergistic, effect (Appendix A). At a fixed ratio of 2:3, the ƩFIC was less than 0.5, and the corresponding point on the isobologram fell much lower than the others, suggesting a strong synergistic relationship between the two compounds.

The total ion chromatograms of fractions #5–#8 revealed a combination of cyclodepsipeptides and an unknown number of additional contaminants, making it impossible to know which compounds are interacting (Appendix A). To better understand this synergy, combination studies were repeated with pure valinomycin and fraction #13 (cPPGs).

Combination studies with valinomycin and fraction #13 also revealed synergy between the two compounds. When plotted as an isobologram, the FIC_50_s of valinomycin and the cPPGs fell below the additivity line and had a concave curve, again suggesting synergy (Figure 9). The fixed ratios of 4:1 and 1:4 had ƩFICs slightly above 0.5, suggesting additivity to weak synergy (Appendix A). Meanwhile the fixed ratios of 3:2 and 2:3 had much lower ƩFICs of 0.37 and 0.43, respectively, suggesting strong synergy. These results reinforce the previous fraction combination studies and strongly suggest that cPPGs synergistically improve the antiplasmodial activity of valinomycin *in vitro*.

## 5. Cytotoxicity Studies

Valinomycin, the cPPGs and their fixed ratios were tested against the Chinese Hamster Ovary (CHO) cell line to determine if the observed antiplasmodial synergy was seen against mammalian cell lines as well.

When tested against the CHO cell line, valinomycin displayed moderate cytotoxicity, with an IC_50_ of 12.40 ± 1.1 µg/mL recorded, and the cPPGs showed no cytotoxicity up to 100 µg/mL (Table 3). This is to be expected from valinomycin, as it is known to display host toxicity [2,38]. These data suggest that the cPPGs selectively target the *P. falciparum* parasite and do not act on mammalian cell systems. Despite this, their antiplasmodial activity is moderate, making them unsuited for further study and development alone [39]. The fixed ratio combinations of valinomycin and cPPGs showed much higher IC_50_s than valinomycin, ranging from 57–65 µg/mL (Table 3). The calculated selectivity indices for the fixed ratio combinations were at least tenfold greater than those of valinomycin and the cPPGs, indicating good host selectivity. This indicates that these combinations are far less toxic than valinomycin alone, and that the synergistic effect observed previously is specific for *P. falciparum* and not mammalian cells.

Valinomycin is a unique and potent antibiotic that is not of clinical use because of its toxic side effects [2]. Combination treatment with synergistic partners is one method to overcome this hurdle, as it requires lower concentrations of the toxic component to reach the same antibiotic effect [40]. This is possibly why valinomycin’s cytotoxicity was reduced by combination with cPPGs.

## 6. Conclusions

In conclusion, in this study we elucidated cPPG ethers, which may act as crown ethers based on molecular dynamics analysis. Further host-guest modelling is required to confirm if these are indeed novel crown ethers. The cPPGs were found to synergistically improve the antiplasmodial activity of valinomycin *in vitro*. This synergy was not observed against the CHO cell line, suggesting that the combinations are selective for *P. falciparum*. By using drug combinations with synergistic partners, the toxic side effects of a drug can be diminished, as lower concentrations are required. This may make valinomycin suitable for use in a clinical setting; however, further study is required to confirm this.

It is surprising that these cPPGs have not been reported before. It is likely that they have appeared in other studies but have been misidentified as PPG or simply did not warrant further study. The existence of these cPPGs must be noted for other bioassay-guided studies. PPG is a common contaminant and, while the cPPGs only displayed moderate antiplasmodial activity, this may still be enough to produce false positive results in other experiments.

## 7. Methods

### 7.1. Fermentation and Extraction

*Streptomyces* strain PR3 was cultured from frozen glycerol stocks (15%, *v*/*v*) in 15 mL of Japan Collection of Microorganisms (JCM) medium #61 (15 g starch, 4 g yeast extract, 0.5 g K_2_HPO_4_, 0.5 g MgSO_4_·7H_2_O, pH 7.4) [41] liquid medium in a 250 mL Erlenmeyer flask. After 3 days of incubation at 30 °C with shaking, this seed culture was used to inoculate 100 mL JCM #61 in a 1 L Erlenmeyer flask. After 3 days of incubation at 30 °C with shaking, this second seed culture was used to inoculate 1 L JCM #61 in a 5 L Erlenmeyer flask. This culture was incubated for 14 days at 30 °C with shaking. Then, 14–16 h prior to extraction, 50 g/L of Amberlite XAD 16-N extraction resin was added to the culture under nonsterile conditions. For extraction, the cell mass and resin were filtered through two coffee filters (size 1 × 4, House of Coffees, Johannesburg, South Africa) and the broth fraction was discarded. Two hundred milliliters of MeOH (>99%, Kimix Chemical and Lab Supplies, Cape Town, South Africa) was added to the filtered cell mass plus resin and agitated on a Labnet^TM^ Orbit 1000 shaker at 140 rpm for 60 min at room temperature (22 °C). The MeOH was then recovered by filtering through two 1 × 4 coffee filters, and the MeOH was discarded. Two hundred milliliters of EtAc (≥98%, Merck, Darmstadt, Germany) was added to the MeOH-extracted cell mass plus resin and agitated on a Labnet^TM^ Orbit 1000 shaker at 140 rpm for 90 min at room temperature. The EtAc was filtered from the cell mass and resin with two 1 × 4 coffee filters and collected in a glass beaker. Another 200 mL of EtAc was added to the filtered cell mass plus resin and agitated on a Labnet^TM^ Orbit 1000 shaker at 140 rpm for 90 min at room temperature. The EtAc was recovered (as before) and combined with the first EtAc wash in a glass beaker. The pooled EtAc fractions were left in a fume hood overnight to dry at room temperature. Each crude extract was redissolved in 5 mL of EtAc and transferred into a clean, pre-weighed plastic benchtop tube (Inqaba Biotec, Pretoria, South Africa).

### 7.2. Fractionation

Crude extracts were initially separated using Phenomenex^®^ Strata^TM^ X-33 µm Reverse Phase C18 SPE cartridges (200 mg, 3 mL) (Separations, Johannesburg, South Africa). A SPEEDISK^®^ 48 manifold was used to wash solvents and dissolved crude extract samples through the C18 cartridges under pressure. The C18 cartridges were equilibrated before use by washing 2 mL MeOH (99.9%, Honeywell, Johannesburg, South Africa) through each cartridge, followed by 2 mL water under pressure. The crude extract sample was dissolved in 2 mL water/MeOH (1:1), then passed through the C18 cartridge under pressure. Two milliliters of solvents of decreasing polarity was washed through the cartridge. The solvents, in order of use, were water/MeOH (1:1), water/MeOH (1:4), MeOH and acetone (99.8%, Honeywell, Johannesburg, South Africa). Each fraction was collected separately and labelled from 1–4.

Fraction #3 was separated by ISOLUTE^®^ silica SPE cartridges (200 mg, 3 mL) (Shimadzu, Roodepoort, South Africa). SPE cartridges were equilibrated by passing 2 mL of MeOH followed by 2 mL of hexane through each cartridge. Fraction #3 was dissolved in 2 mL hexane/EtAc (9:1) and washed through the silica cartridges by gravity. After adding the sample, solvents of increasing polarity were passed through the silica cartridges. The list of solvents in order of use was: hexane (≥97%, Sigma-Aldrich, Johannesburg, South Africa), hexane/EtAc (8:1), hexane/EtAc (7:1), hexane/EtAc (6:1), hexane/EtAc (5:1), hexane/EtAc (4:1), hexane/EtAc (3:1), hexane/EtAc (2:1), hexane/EtAc (1:1), EtAc, EtAc/MeOH (4:1), EtAc/MeOH (2:1) and MeOH. When using the silica cartridges, the sorbent was not allowed to dry, as this alters the retention of compounds in the subsequent wash steps. Therefore, 2.5 mL of solvent was added in each step, and approximately 0.5 mL of each solvent was left behind (i.e., only 2 mL of each fraction was collected).

### 7.3. P. falciparum, NF54 Parasite Lactate Dehydrogenase Assay

Antiplasmodial testing was done against the drug-sensitive strain of *P. falciparum*, NF54 (Patient Line E), accession number MRA-1000, obtained from the Malaria Research and Reference Reagent Resource (MR4) depository. *P. falciparum* was continuously cultured at 37 °C in human O^+^ erythrocytes using a modified version of the method described by Trager and Jensen [42]. The culture medium was composed of 10.4 g/L RPMI 1640 (with glutamine and without NaHCO_3_), 4 g/L glucose, 6 g/L Hepes buffer, 0.088 g/L hypoxanthine, 5 g/L albumax, and 102 mL/L (0.05 g/L) gentamicin (Sigma-Aldrich, Johannesburg, South Africa) at pH 7.4. Human blood, donated by anonymous donors, was obtained from the Western Cape Blood Service (Cape Town, South Africa). The half-maximal inhibitory concentration (IC_50_) of each fraction and crude extract was determined using a modified version of the parasite lactate dehydrogenase (pLDH) assay described by Makler et al. [43]. The assay was performed in quadruplicate with a 10-point dose-response curve (two fold serial dilutions) over the concentration range 5.00–0.010 µg/mL. The data were analyzed by nonlinear regression analysis using the GraphPad PRISM version 4.00 program to determine the IC_50_ of each compound. Chloroquine and artesunate (Sigma-Aldrich, Johannesburg, South Africa) were used as reference standards. Valinomycin (≥98% purity, Merck^®^, Darmstadt, Germany) was tested in the same manner, but at a concentration range of 100–0.01 ng/mL.

### 7.4. Spectroscopic Analysis

HPLC–HRESIMS was obtained using an AB Sciex^®^ X500R QTOF coupled to an AB Sciex^®^ Exion LC system. Spectral data were obtained using information dependent acquisition (IDA) at a mass range of 50–1300 Da. All methods, batches and data were processed using OS Sciex^®^ v1.2. The declustering potential was 80 V, the curtain gas (N_2_) was at 25 pounds per square inch (psi), the ion spray voltage was 5500 V, and the source temperature was 450 °C. Ion source gases 1 and 2 were at 45 and 55 psi, respectively. The collision energy was 10 eV for the MS scans and 20–50 eV for MS/MS scans. The IDA intensity threshold was 50 cycles per second. The aqueous mobile phase used was 1 mM ammonium formate in water, and the organic mobile phase was MeOH/0.5% formic acid. The method used a gradient starting at 2% organic and ending at 98% organic phase, with a flow rate of 600 µL/min and a run time of 18 min used. A Kinetex^®^ Evo C18 LC column (5 µm, 100 Å, 50 mm × 2.1 mm) with a column protector was used. All solvents were sonicated for 10 min before use to remove bubbles.

One and two dimensional NMR spectra were obtained on a BRUKER Ascend 600 (Bruker, Billerica, MA, USA) using a Prodigy cryoprobe at 600 and 150 MHz for ^1^H and ^13^C nuclei, respectively. Samples were dissolved and analyzed in CD_3_OD (_H 3.30, _C 49.0).

Chromatograms were visualized using MZMine2.0 [44].

### 7.5. Global Natural Products Social Molecular Networking

Raw HRESIMS data were converted to mzXML format using ProteoWizard tool MSconvert (version 3.0.10051, Vanderbilt University, Nashville, TN, USA) [45]. The converted files were uploaded to the GNPS molecular networking server and analyzed by the molecular networking workflow [17]. A molecular network was created using the online workflow (https://ccms-ucsd.github.io/GNPSDocumentation/, accessed on 15 July 2021) on the GNPS website (http://gnps.ucsd.edu, accessed on 15 July 2021). The data were filtered by removing all MS/MS fragment ions within ± 17 Da of the precursor m/z. MS/MS spectra were window filtered by choosing only the top 6 fragment ions in the ±50 Da window throughout the spectrum. The precursor ion mass tolerance was set to 2.0 Da with a MS/MS fragment ion tolerance of 0.5 Da. A network was then created where edges were filtered to have a cosine score above 0.7 and more than 6 matched peaks. Further, edges between two nodes were kept in the network if, and only if, each of the nodes appeared in each other’s respective top 10 most similar nodes. Finally, the maximum size of a molecular family was set to 100, and the lowest scoring edges were removed from molecular families until the molecular family size was below this threshold. The spectra in the network were then searched against the GNPS spectral libraries. The library spectra were filtered in the same manner as the input data. All matches kept between network spectra and library spectra were required to have a score above 0.7 and at least 6 matched peaks. The results were visualized using Cytoscape v3.7.2 [46].

### 7.6. Molecular Dynamics Setup and Analysis

System Setup and Molecular Dynamics: The diastereomer (with all R chiral centers) of a selected cPPG (compound 870 with 15 oxygen atoms) was initially built using AmberTools [47] and VMD [29]. The PDB model was then rebuilt and parameterized using CHARMM-GUI’s [48] ligand reader and modeler to generate CHARMM-compatible topology and parameters. The Multicomponent assembler of CHARMM-GUI was used to build solvated systems of the macrocycle with each salt (at 0.15 M) at 303.15 K. Each system is a cubic waterbox (initial dimensions of 5.0 nm 5.0 nm 5.0 nm). The salts used were NaCl, KCl, and NH_4_Cl. OpenMM inputs were generated. The CHARMM 36 Forcefield was used. The OpenMM software [49] was then utilized for molecular dynamics. A short minimization (5000 steps) was followed by a brief equilibration stage (125 ps with a time step of 0.001 ps). Particle Mesh Ewald Electrostatics was used, along with force-switching for van der Waals interactions (switch on from 1.0 nm and switch off at 1.2 nm). NPT Production dynamics totaling 200 ns were carried out for each system, with a time step of 0.002 ps used. TTclust software [30] was used to cluster and determine the conformations of the macrocycle.

MDAnalysis [28] was used to analyze trajectories, align structures, and calculate properties of the macrocycle. For example, the root-mean square deviation (RMSD), root-mean square fluctuation (RMSF), and radius of gyration (R_gyr_) were used to investigate the average structural fluctuations; meanwhile, a radial distribution function (RDF) was used to measure the interaction between the macrocycle and NaCl, NH_4_Cl and KCl in a water box. A radial distribution function (RDF) using MDAnalysis was used to measure the normalized probability distribution of any neighboring cation within a certain radius of the macrocycle. This was further refined by selecting only the oxygen atoms.
(1)$$g_{\mathrm{ab}}\left(r\right)={\left(N_{a}N_{b}\right)}^{-1}\overset{N_{a}}{\underset{i=1}{\sum }}\overset{N_{b}}{\underset{j=1}{\sum }}\langle \delta \left(\left|r_{i}-r_{j}\right|-r\right)\rangle$$

### 7.7. Cytotoxicity Assays

Cytotoxicity assays were conducted against the Chinese Hamster Ovary (CHO) cell line (ATCC CCL-61, strain CHO-K1) by the H3D parasitology unit at the University of Cape Town, South Africa. CHO cells were cultured in a medium consisting of 10% foetal calf serum (FCS, Celtic Molecular Diagnostics, Mowbray, South Africa), 45% Dulbecos Modified Eagles Medium (DMEM, Highveld Biologicals, Lyndhurst, South Africa) and 45% HAMS-F12 medium (1:1; Sigma, St. Louis, MO, USA). Cells were grown at 37 °C in a humidified atmosphere of 5% CO_2_ and maintained by passage. Cells lines were seeded in 96-well-microtitre plates at 10^4^ cells/well in cell medium. Once seeded, cells were incubated at 37 °C for 24 h before the test samples were added. Valinomycin, fraction #13 and each fixed ratio drug combination were tested in triplicate at six concentration points (100, 10, 1, 0.1, 0.01 and 0.001 µg/mL). Emetine was used as the reference standard for cytotoxicity. Cell viability was determined using the colorimetric 3-(4,5-dimethylthiazol-2-yl)-2,5-diphenyl tetrazolium bromide (MTT) assay as described by Mosmann [50].

One hundred microliters of dimethyl sulfoxide (DMSO, Sigma-Aldrich, Johannesburg, South Africa) was added to dissolve the MTT formazan derivatives, and the plates were gently agitated for 2 min to ensure homogenous mixtures. Absorbance was measured at 540 nm using a Turner BioSystems, Inc. Modulus^TM^ II Microplate Reader. The data were analyzed by nonlinear regression analysis using the GraphPad PRISM version 4.00 program to determine the IC_50_ of each compound.

### 7.8. Fixed-Ratio Isobologram

Combination studies were conducted using a modified version of the method described by Fivelman et al. (2004) [35]. Drug stock solutions were made up such that their approximate IC_50_s were obtained after 3–5 two-fold dilutions. The drug stocks were then mixed to make 6 different volumetric ratios to obtain a range of concentrations (Table 4 and Table 5). The FIC_50_ and ƩFIC were calculated as described in Equation (2) [37]. Results were displayed using the GraphPad PRISM version 4.00 program.
(2)∑FIC=FICA+FICB
where $FICA=\frac{\left[A\right]}{IC50A}$ and $FICB=\frac{\left[B\right]}{IC50\;B}$.

## 8. Data Availability

### 8.1. Molecular Networks

Fraction #5: https://gnps.ucsd.edu/ProteoSAFe/status.jsp?task=c704c63bf4a94d6da6f0b1d6df0628b9, 15 July 2021.

Fraction #6: https://gnps.ucsd.edu/ProteoSAFe/status.jsp?task=29dc421a531c44a094432d0ba98a9ad1, 15 July 2021.

Fraction #7: https://gnps.ucsd.edu/ProteoSAFe/status.jsp?task=bdbcd8ee65fd4f8da2dd48257f0617f6, 15 July 2021.

Fraction #8: https://gnps.ucsd.edu/ProteoSAFe/status.jsp?task=c4d5c26d89c54972828f47265c85bfde, 15 July 2021.

Fraction #13: https://gnps.ucsd.edu/ProteoSAFe/status.jsp?task=5997444ae6184a1da3aa7f3e05852897, 15 July 2021.

### 8.2. Dereplicator

Fraction #5: https://gnps.ucsd.edu/ProteoSAFe/status.jsp?task=599bd623e108400283aa4548b6e0d251, 15 July 2021.

Fraction #6: https://gnps.ucsd.edu/ProteoSAFe/status.jsp?task=235c8621fac744aa9c03182f69d34786, 15 July 2021.

Fraction #7: https://gnps.ucsd.edu/ProteoSAFe/status.jsp?task=965eb43b8f6242cabbd91bfc879491ee, 15 July 2021.

Fraction #8: https://gnps.ucsd.edu/ProteoSAFe/status.jsp?task=57899acaafef45cf9eac002cd2d95375, 15 July 2021.

### 8.3. Varquest

Fraction #5: https://gnps.ucsd.edu/ProteoSAFe/status.jsp?task=37d3dc516c524631b2e83cd6cdbe7280, 15 July 2021.

Fraction #6: https://gnps.ucsd.edu/ProteoSAFe/status.jsp?task=f7e8e904b2224f34b0fa65d9563ce6a1, 15 July 2021.

Fraction #7: https://gnps.ucsd.edu/ProteoSAFe/status.jsp?task=07bf0532f0634ad3aca6f3cfa82fca3f, 15 July 2021.

Fraction #8: https://gnps.ucsd.edu/ProteoSAFe/status.jsp?task=cd4e4382851e4e6382762201d4a3945b, 15 July 2021.

## Figures and Tables

**Figure 1 molecules-26-07494-f001:**
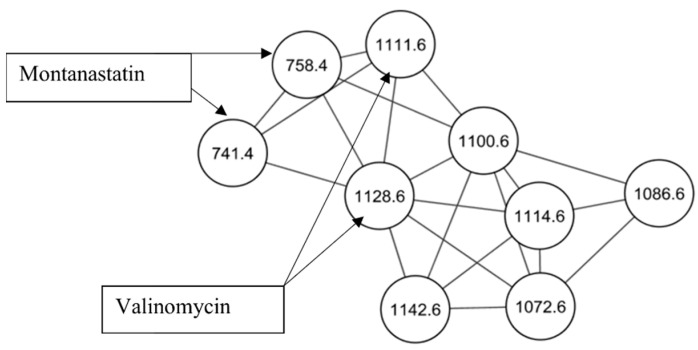
Representative molecular network of the cyclodepsipeptides from fraction #6.

**Figure 2 molecules-26-07494-f002:**
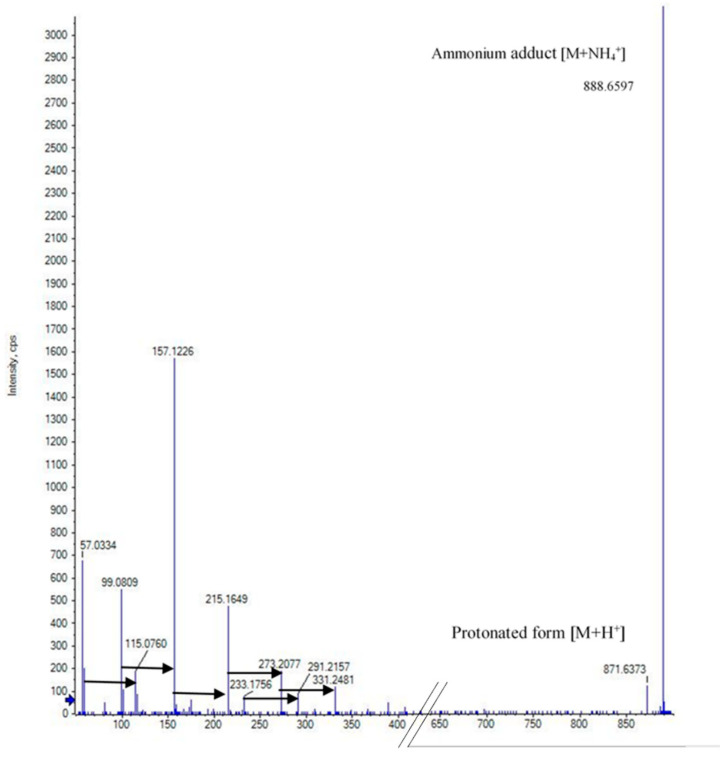
The fragmentation mass spectra of compound 870. Black arrows represent a mass shift of 58.04 Da, which indicates a repeating unit.

**Figure 3 molecules-26-07494-f003:**
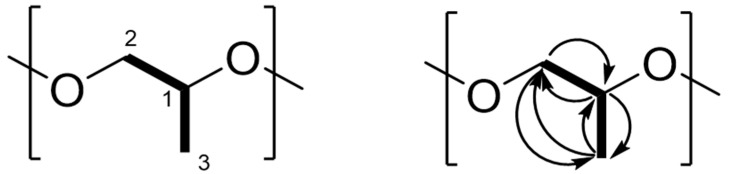
Structure and COSY (▬) and HMBC (→) correlations for the C_3_H_6_O monomer.

**Figure 4 molecules-26-07494-f004:**
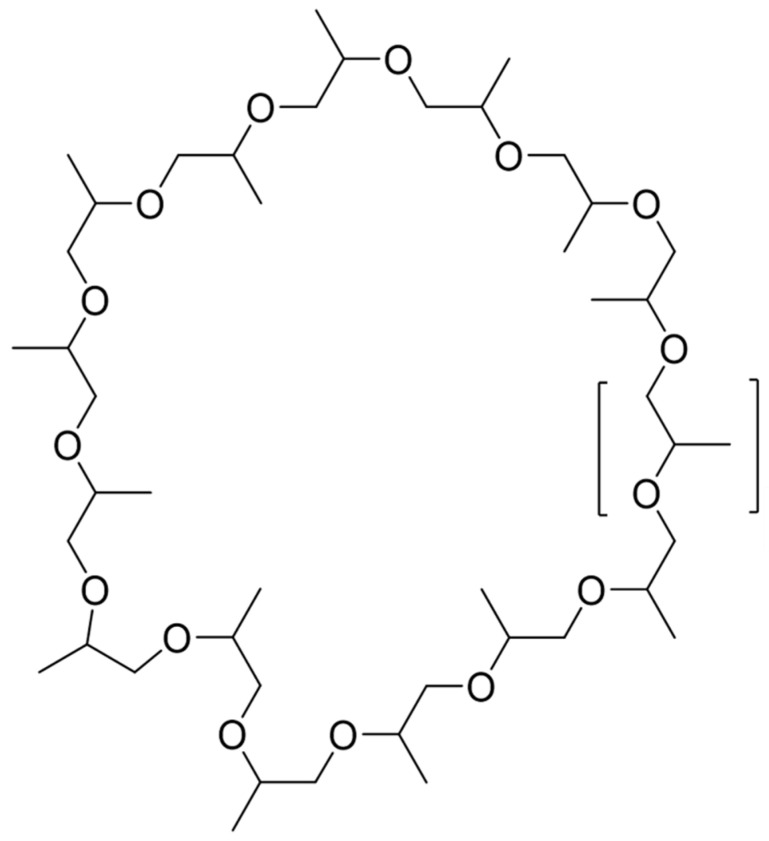
Proposed structure of the cyclic polypropylene glycol. The square bracket indicates the proposed propylene glycol monomer.

**Figure 5 molecules-26-07494-f005:**
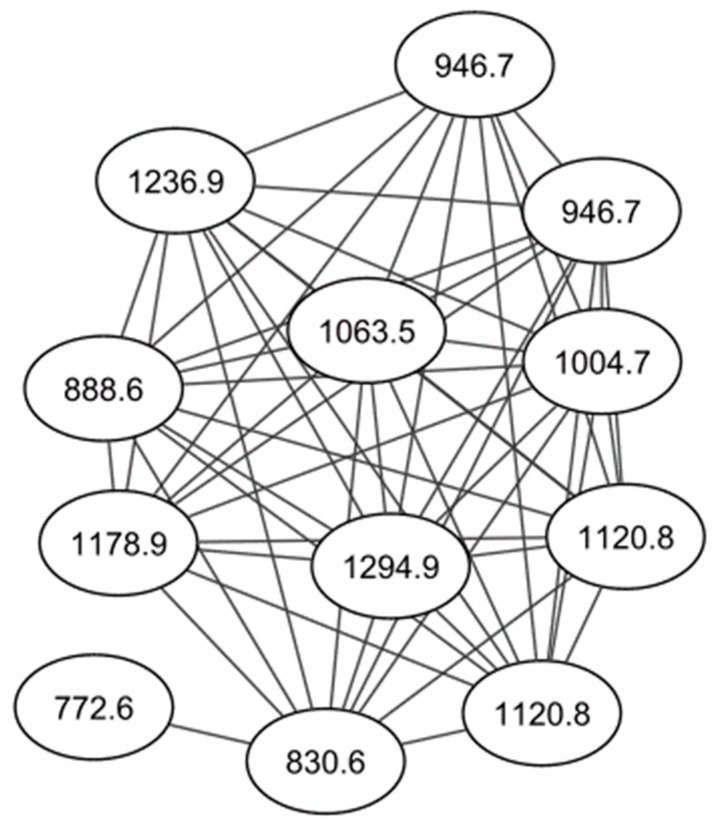
A molecular network of the cyclic polypropylene glycols from fraction #13.

**Figure 6 molecules-26-07494-f006:**
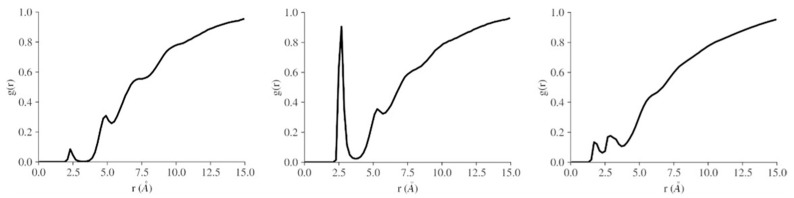
RDFs of the oxygens of the macrocycle (compound 870) with sodium, potassium and ammonium cations, respectively.

**Figure 7 molecules-26-07494-f007:**
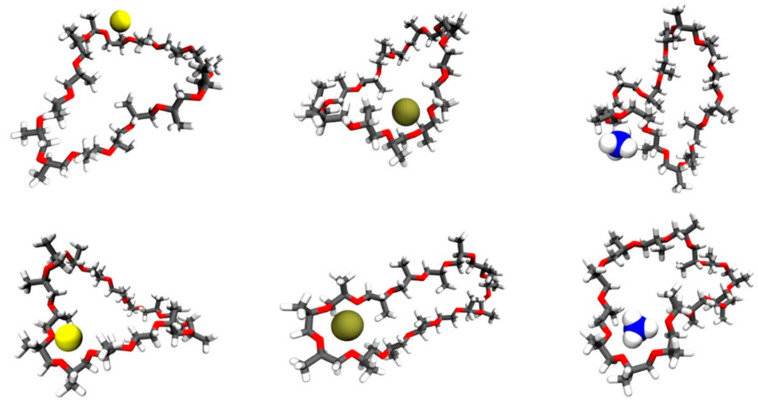
Representations of interesting bound states of each ion and compound 870 observed during molecular dynamics simulations. Ions are represented as space-filling spheres, while the macrocycle is represented as licorice. Sodium is shaded yellow; Potassium brown and the nitrogen of ammonium is shaded blue. The first row illustrates bidentate motifs, while the second row illustrates crown ether motifs.

**Figure 8 molecules-26-07494-f008:**
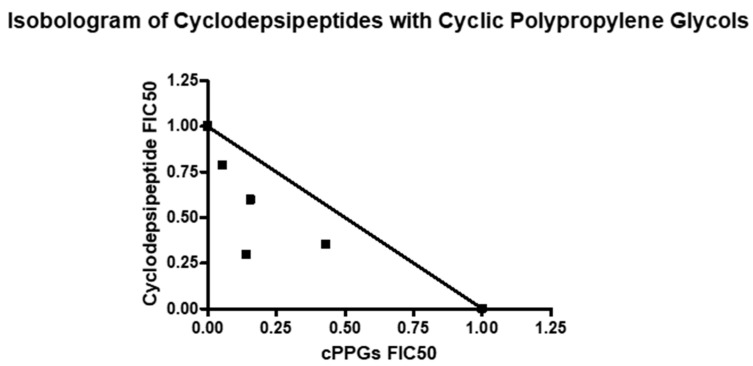
Isobologram of cyclodepsipeptides (fractions #6) and cyclic polypropylene glycols (fraction #13).

**Figure 9 molecules-26-07494-f009:**
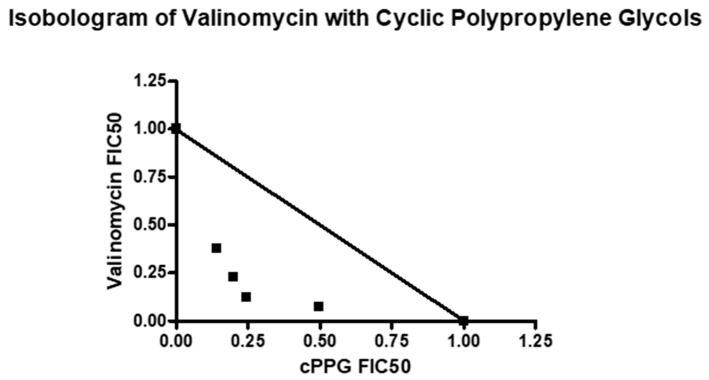
Isobologram of valinomycin and cyclic polypropylene glycols (fraction #13).

**Table 1 molecules-26-07494-t001:** Mean antiplasmodial activity of *Streptomyces* strain PR3 crude extract and fractions against *P. falciparum*, NF54, *N* = 3 biological repeats with 4 technical repeats.

Sample	Antiplasmodial Activity against *P. falciparum,* NF54 IC_50_ (ng/mL)
Crude Extract	497 ± 14
Reverse Phase SPE Fractions	
#1 Water	>5000
#2 Water/MeOH (1:1)	>5000
#3 MeOH	40.0 ± 1.5
Normal Phase SPE Fractions	
#4 Hexane	>5000
#5 Hexane/EtAc (8:1)	150.0 ± 53
#6 Hexane/EtAc (7:1)	31.3 ± 4.2
#7 Hexane/EtAc (6:1)	63.5 ± 17.0
#8 Hexane/EtAc (5:1)	287.0 ± 14.0
#9 Hexane/EtAc (4:1)	>5000
#10 Hexane/EtAc (3:1)	>5000
#11 Hexane/EtAc (2:1)	>5000
#12 Hexane/EtAc (1:1)	>5000
#13 EtAc	2105.0 ± 146.0
#14 EtAc/MeOH (4:1)	>5000
#15 EtAc/MeOH (2:1)	>5000
#16 MeOH	>5000
Positive Controls	
Chloroquine	7.1 ± 2.1
Artemisinin	3.3 ± 1.3

**Table 2 molecules-26-07494-t002:** ^13^C, ^1^H, COSY and HMBC data for the cyclics polypropylene glycol monomer.

Position	^13^C	^1^H mult	^13^C-^1^H COSY	^13^C-^1^H HMBC
1	75.3, CH	3.63 m	2,3	2,3
2	72.87, CH_2_	3.55 3.48 m	1	1,3
3	16.22, CH_3_	1.16 m	2	1,2

**Table 3 molecules-26-07494-t003:** Mean cytotoxicity of valinomycin, cPPGs (fraction #13) and their fixed ratios against the Chinese Hamster Ovary cell line. *N* = 2 with 3 technical repeats.

Sample	Cytotoxicity against CHO, IC_50_ (µg/mL)	Antiplasmodial Activity against *P. falciparum*, NF54 IC_50_ (ng/mL)	Selectivity Index (SI) *
Valinomycin	12.40 ± 1.1	3.75 ± 0.77	3306
Fixed Ratio 4:1	60.80 ± 6.3	1.86 ± 0.05	32,688
Fixed Ratio 3:2	57.88 ± 6.1	0.90 ± 0.20	64,311
Fixed Ratio 2:3	65.25 ± 1.3	0.75 ± 0.08	87,000
Fixed Ratio 1:4	58.43 ± 6.1	0.53 ± 0.1	110,245
Cyclic Polpropylene Glycols (Fraction #13)	>100 ± ND	1792 ± 547	>55
Emetine	0.03 ± 0.002	ND	ND

* SI = CHO IC_50_/NF54 IC_50_; ND = Not determined.

**Table 4 molecules-26-07494-t004:** Fixed ratios of cyclodepsipeptides (fraction #6) and cPPGs (fraction #13).

Ratio Number	Fixed Ratio	Cyclodepsipeptides Concentration (µg/mL)	Cyclic Polypropylene Glycols Concentration (µg/mL)
1	5:0	1.6	0
2	4:1	1.28	8
3	3:2	0.96	16
4	2:3	0.60	24
5	1:4	0.32	32
6	0:5	0	40

**Table 5 molecules-26-07494-t005:** Fixed ratios of valinomycin and cPPGs (fraction #13).

Ratio Number	Fixed Ratio	Valinomycin Concentration (ng/mL)	Cyclic Polypropylene Glycols Concentration (µg/mL)
1	5:0	160	0
2	4:1	128	8
3	3:2	96	16
4	2:3	64	24
5	1:4	32	32
6	0:5	0	40

## Data Availability

All data are available upon request.

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
