# Peer review of "Discovery of Novel Cyclic Ethers with Synergistic Antiplasmodial Activity in Combination with Valinomycin"

_molecules, 2021, doi:10.3390/molecules26247494_

Round 1

Reviewer 1 Report

Comments to the Author:

This is the review of the manuscript entitled Discovery of Novel Crown Ethers with Synergistic Antiplasmodial Activity in Combination with Valinomycin

The authors of this paper performed the bioassay guided fractionation of Streptomyces strain PR3 liquid culture and identified fractions with antimalarial activity. One of the active fractions contained a group of cyclodepsipeptides including valinomycin, while in the other a novel class of cyclic ethers was identified and elucidated. Structure analysis and molecular dynamic analysis identified these cyclic ethers as a novel class of crown ethers. The synergy between crown ethers and valinomycin was also demonstrated.

In comparison to previous version of the manuscript the authors did a considerable effort to clarify some imprecisions, added new experiments, draw the adequate conclusions and significantly improved the manuscript. There are still some minor points they should pay attention to:

INTRODUCTION:

Line 39-40: However, most ionophores are barred from being used as chemotherapeutic agents in humans due to their lack of cytotoxicity.

 RESULTS AND DISCUSSION

Line 315-318: This is to be expected from valinomycin, as it is known to display host toxicity. This data suggests that the cPPGs are selectively targeting the P. falciparum parasite and are not acting on mammalian cell systems. The lack of cytotoxicity from the cPPGs is unique among crown ethers, which generally display poor host selectivity.

I suggest the authors to rephrase a bitt these statements because it should be clear if

Ionophores in general show the lack of cytotoxicity, they have anticancer potential (cytotoxic activity against cancer, not healthy cells) or crown ethers do show cytotoxicity but they also have poor host selectivity?

I realize that the authors are referring to findings from published articles but they should be careful not to oversimplify conclusions and hence sound a bit confusing.  

REFERENCES

 All references should be formatted in the same manner (according too the Instructions for Authors). If I am not mistaking, the Reference List and Citations Style Guide for MDPI Journals says:

Reference citation numbers should be placed in square brackets, i.e. [ ], and placed inside the punctuation, for example [4].

For documents co-authored by a large number of persons (more than 10 authors), you can either cite all authors, or cite the first ten authors, then add a semicolon and add ‘et al.’ at the end.

.As an example:

Díaz, D.D.; Converso, A.; Sharpless, K.B.; Finn, M.G. 2,6-Dichloro-9-thiabicyclo[3.3.1]nonane: Multigram Display of Azide and Cyanide Components on a Versatile Scaffold. Molecules 2006,11, 212–218, doi:10.3390/11040212.

The authors should go through all references and make the necessary corrections.

Author Response

We thank the reviewer for their input and comments. Attached is a response to each comment.

Reviewer 2 Report

 This manuscript by the group of Watson revised a previous version “entitled Discovery of Novel Crown Ethers with Synergistic Antiplasmo-dial Activity in Combination with Valinomycin”. The key revision in this manuscript would be the support for the generation of cyclic oligo polypropylene glycol (cPPG). Though authors added further experiments and computations, I feel the key concept have not been supported thoroughly. Most critically, the generation and its host-guest interactions of cPPG would be weak. First, the mass spectrum peak of cPPG at m/z of 871.6373 should involve isotope pattern that is unique to the molecular compositions. The authors should carefully analyze the data. Second, in the 1H NMR spectrum of the compound in Figure S6, integration values should be documented. In addition, the unassigned peaks should be re-analyzed and assigned carefully. Too many peaks left unassigned. Third, why the authors sticks to “crown ethers” ? I think the authors tried to show that the obtained molecules could be “cyclic” PPG. Though authors computationally analyzed the cPPG, the interaction between oligo ether moieties and cationic species are inherent and not surprising. Unless the cyclic conformations made a decisive contribution to host-guest chemistry, the authors should not claim that the cyclic compounds are "crown ethers". I think authors could modify the title in the way that the obtained data reflects the effects of cyclic PPG but NOT crown ethers. All in all, I think the manuscript should limit to the argument that their current data could support. Below comments are minor remarks on this manuscript.

  1. Figures 8 and 9 must be improved. The current figures are unmature for publications.

Author Response

We thank the reviewer for their inputs and comments. Attached is a response to their review.

Round 2

Reviewer 2 Report

The revisions made were sufficient to support their arguments.

This manuscript is a resubmission of an earlier submission. The following is a list of the peer review reports and author responses from that submission.

Round 1

Reviewer 1 Report

The manuscript by Watson and their co-workers has tried to present medical applications of cyclic oligoethers in the combination with valinomycin. Most strikingly, I do not find a rational connection between cyclic oligoethers and their effects on the medical applications. In the introduction, the authors mentioned about ionophores. However, NO experimental/theoretical insights into molecule-binding properties were presented. In case the authors wish to claim their cyclic oligoether as a "crown ether", they must evaluate molecule-binding properties. However, I think this type of molecule can not behave as a good ionophores because the cyclic ring is so large that the pre-organization of crown-like structure do not fit into target molecules. Therefore, the authors should carefully verify the cyclic ethers. In addition, the NMR characterization of the product is not convincing. The authors should reconsider the research design. Last but not least, I can not find a rational reason to combine cyclic oligoether and valinomycin. All in all, I think this manuscript is too premature for publication and I thus would not support the publication of this manuscript. In addition, following issues should be handled prior to publication.

1) The figure quality is too low for publication. Please carefully prepare publication-grade figures.

2) The authors mentioned the cyclic oligoether as a [150]crown-15-ether. This is not true. Only if the compound shows the crown-ether like behaviour, the compound should be 50-crown-15-ether.

Reviewer 2 Report

This is the review of the manuscript entitled Discovery of Novel Crown Ethers with Synergistic Antiplasmodial Activity in Combination with Valinomycin

The authors of this paper performed the bioassay guided fractionation of Streptomyces strain PR3 liquid culture and identified several fractions with antimalarial activity. The first active fraction contained a cluster of cyclodepsipeptides (including valinomycin and montanastatin) and in the second active fraction the authors identified a novel class of 18 crown ethers. The synergy between crown ethers and valinomycin was also demonstrated.

I would like to point out some of irregularities that I have noticed during reweaving process.

ABSTRACT:

Line 16: During the bioassay guided fractionation of Streptomyces strain PR3?

This sounds a bit imprecise. Actually, the antimalarial activity was detected in crude extract of Streptomyces strain PR3?

 Line 21: The IC50 of valinomycin against P. falciparum NF54 improved by 4-5-fold, from 21 3.75 ± 0.77 ng/mL to 0.90 ± 0.2 ng/mL and 0.75 ± 0.08 ng/mL when dosed by itself and in the fixed 22 ratios of 3:2 and 2:3 (valinomycin to crown ethers), respectively.

This sentence sounds confusing.

Suggestion: The IC50 of valinomycin against P. falciparum NF54 improved by 4-5-fold when valinomycin was combined with crown ethers. Precisely, it was improved from 3.75 ± 0.77 ng/mL to 0.90 ± 0.2 ng/mL and 0.75 ± 0.08 ng/mL when dosed in the fixed ratios of 3:2 and 2:3 of valinomycin to crown ethers, respectively.

 RESULTS AND DISCUSSION

Line 64: Streptomyces strain PR3 was identified as a producer of potent antiplasmodial compounds during a series of experiments to screen filamentous actinobacteria.   

Where were these screening experiments described? Reference? Assay for antiplasmodial activity?

Line 96-97: As these analogues possess the same core ring structure as valinomycin, they are likely acting by a similar mechanism of action.

If the authors want to potentiate the mechanism of action of cyclodepsipeptides present in the fractions #5-#8 then they should perform the additional experiments and discuss the result thoroughly, comparing them with the literature data.

Line 154-156: Mass spectrometric analysis of the growth medium and extraction resin revealed the presence of [150]crown-15. It also revealed that washing the XAD16-N resin with EtAc and water, produced a sample containing [150]crown-15.

Line 161-163: The exact mechanism of cyclization was not elucidated, but this information suggests that, following the addition of the XAD16-N resin prior to extraction, PPG oligomers leached and cyclized to form [150]crown-15 in the aqueous culture broth.

These sentences are confusing. If the [150]crown-15 originates from XAD16-N resin what is theexplanation for their presence in the growth medium? They could be expected in the culture broth but why in the growth medium? Why the mass spectrometric analysis of the growth medium and extraction resin  is not presented?

Line 175-258: Combination Studies

In my opinion, this part of the manuscript is written in too many details.  The FIC index analysis is well described in the literature and the authors stated the adequate references for this methodology. Finally, these results could be presented in one way only, for example with tables  (without isobolograms) and the appropriate conclusion that synergistic effect was detected when fraction #13 was combined either with fraction #6 or pure valinomycin.

Line 259-275: Cytotoxicity Studies

Regarding cytotoxicity studies, my opinion is that the authors should discuss more about the selectivity of cyclic polypropylene glycol oligomers for P. falciparum, comparing to CHO cell line.

Line 276-297: Conclusion

In my opinion, this part is too long, with too many general statements (require more extensive study, it is possible that…, may have alternative applications…, it is likely that they appeared….). Main conclusions should be made on the basis of results presented in this manuscript.

REFERENCES

 The references should be formatted in the same manner (according too the Instructions for Authors). There are some incorrect/missing data:

Line 454: 8. Maron, M. I. et al. Maduramicin Rapidly Eliminates Malaria Parasites and Potentiates the Gametocy-454 tocidal Activity of the Pyrazoleamide PA21A050. Antimicrob. Agents Chemother. 60, 1492–1499 (2016).

In the PubMed they say that this manuscript should be cited as

Maron, M. I. et al. Maduramicin Rapidly Eliminates Malaria Parasites and Potentiates the Gametocytocidal Activity of the Pyrazoleamide PA21A050. Antimicrob Agents Chemother. 2015 Dec 28;60(3):1492-9. doi: 10.1128/AAC.01928-15. PMID: 26711768; PMCID: PMC4775975.

Line 502: 32. Adovelande, J. & Schrével, J. Carboxylic ionophores in malaria chemotherapy: The effects of monensin and nigericin on and. Life Sci. 59, PL309–PL315 (1996).

Should be

Adovelande J, Schrével J. Carboxylic ionophores in malaria chemotherapy: the effects of monensin and nigericin on Plasmodium falciparum in vitro and Plasmodium vinckei petteri in vivo. Life Sci. 1996;59(20):PL309-15. doi: 10.1016/s0024-3205(96)00514-0. PMID: 8890952.

Line 502: 41. Dowler, M. G. Personal Communication. (1995). ?